# Regulation of Fungal Morphogenesis and Pathogenicity of *Aspergillus flavus* by Hexokinase AfHxk1 through Its Domain Hexokinase_2

**DOI:** 10.3390/jof9111077

**Published:** 2023-11-04

**Authors:** Zongting Huang, Dandan Wu, Sile Yang, Wangzhuo Fu, Dongmei Ma, Yanfang Yao, Hong Lin, Jun Yuan, Yanling Yang, Zhenhong Zhuang

**Affiliations:** 1Key Laboratory of Pathogenic Fungi and Mycotoxins of Fujian Province, Key Laboratory of Biopesticide and Chemical Biology of Education Ministry, Proteomic Research Center, School of Life Sciences, Fujian Agriculture and Forestry University, Fuzhou 350002, China; 13877558694@163.com (Z.H.); kikimra@163.com (D.W.); mbt1709446@xmu.edu.my (S.Y.); fuwangzhuo@foxmail.com (W.F.); 5220543002@fafu.edu.cn (Y.Y.); linhong@m.fafu.edu.cn (H.L.); yjmail2008@126.com (J.Y.); 13635287532@163.com (Y.Y.); 2College of Animal Sciences (College of Bee Science), Fujian Agriculture and Forestry University, Fuzhou 350002, China; 13945932275@163.com

**Keywords:** *Aspergillus flavus*, AfHxk1, AFB_1_, crop kernels, *Galleria mellonella*

## Abstract

As a filamentous pathogenic fungus with high-yield of aflatoxin B_1_, *Aspergillus flavus* is commonly found in various agricultural products. It is crucial to develop effective strategies aimed at the prevention of the contamination of *A. flavus* and aflatoxin. Hexokinase AfHxk1 is a critical enzyme in fungal glucose metabolism. However, the role of AfHxk1 in *A. flavus* development, aflatoxin biosynthesis, and virulence has not yet been explored. In this study, *afHxk1* gene deletion mutant (Δ*afHxk1*), complementary strain (Com-*afHxk1*), and the domain deletion strains (*afHxk1*^ΔD1^ and *afHxk1*^ΔD2^) were constructed by homologous recombination. Phenotype study and RT-qPCR revealed that AfHxk1 upregulates mycelium growth and spore and sclerotia formation, but downregulates AFB_1_ biosynthesis through related classical signaling pathways. Invading models and environmental stress analysis revealed that through involvement in carbon source utilization, conidia germination, and the sensitivity response of *A. flavus* to a series of environmental stresses, AfHxk1 deeply participates in the regulation of pathogenicity of *A. flavus* to crop kernels and *Galleria mellonella* larvae. The construction of domain deletion strains, *afHxk1*^ΔD1^ and *afHxk1*^ΔD2^, further revealed that AfHxk1 regulates the morphogenesis, mycotoxin biosynthesis, and the fungal pathogenicity mainly through its domain, Hexokinase_2. The results of this study revealed the biological role of AfHxk1 in *Aspergillus* spp., and might provide a novel potential target for the early control of the contamination of *A. flavus*.

## 1. Introduction

*Aspergillus flavus* is a worldwide-distributed soil-borne saprophytic fungus that thrives by decomposing organic matter for its growth and survival [1]. One of the major concerns associated with this fungus is its capability to produce aflatoxins (AFs), a class of mycotoxins that contaminate common agricultural products such as cereals, oilseeds, spices, and nuts [2]. Especially, aflatoxin B1 (AFB_1_) is widely recognized as one of the most toxic known substances, with potent carcinogenic effects [3]. These aflatoxins exhibit remarkable heat resistance, retaining their toxicity even after ingested by poultry and livestock [4]. Consequently, when dairy products, meats, and eggs derived from these animals enter the market and are consumed, aflatoxins can directly or indirectly impact human health [5]. The suspected carcinogenic, mutagenic, and immunosuppressive properties of aflatoxins have garnered significant attention since the 1960s [3]. Alarming reports indicated that, in recent years, certain regions have witnessed aflatoxin concentrations exceeding 1000 µg/kg in the aforementioned food items [6]. Furthermore, extensive research has established a strong correlation between these fungal toxins and liver cell damage, potentially leading to fatality and affecting other organs as well [7]. Multiple factors, including climate, environmental conditions, inadequate production practices, and improper storage methods, exacerbate the adverse effects of *A. flavus* on both the economic and health sectors of society [5]. Therefore, addressing the issue of *A. flavus* contamination remains an urgent and critical task.

Current methods to remove aflatoxins are categorized into Pre-Harvest (mainly through biological control agents) and Post-Harvest (sorting, dehulling, milling, adsorbents, acids, etc.) approaches [6,8,9,10,11]. However, existing methods have limitations in terms of efficiency and cost. Currently, there are a growing diversity of biocontrol strategies for the control of aflatoxins based on biological methods [6]. The development of molecular genetics techniques has provided enormous potential for further improvement and optimization of these strategies. Consequently, deeply conducting research on the regulatory mechanism of aflatoxins biosynthesis can lay a foundation for the prevention and management of aflatoxin contamination in future food and feed supplies.

In recent years, researches on *A. flavus* have experienced significant growth, primarily driven by advancements in genetics and molecular biology [12]. The primary focus of these research studies has been to understand the regulatory mechanism of toxin production and pathogenicity [12]. Numerous studies have elucidated the intricate process by which aflatoxins are synthesized through a series of complex enzymatic cascade reactions. These reactions are carried out by 30 genes situated on chromosome 3 [13]. Additionally, central regulatory factors like BrlA→AbaA→WetA, developmental activators such as FlbA, VeA-VelB-LaeA, and VelD, as well as other asexual developmental regulators including AtfA and AtfB, play crucial roles in various biological processes of *A. flavus* [14,15,16,17,18,19]. These encompass the molecular regulatory mechanism of toxin production, normal growth, and sexual reproduction, in which there are still a large number of unknown mechanisms that need to be explored urgently [12,20].

Hexokinases (HKs) are highly conserved enzymes that play a vital role in the initial step of glucose metabolism. They orchestrate a multitude of glucose breakdown and synthesis pathways, including glycolysis, glycogen synthesis, the pentose phosphate pathway, and hexosamine biosynthesis pathways [21,22]. In fungi, hexokinase is identified as a key participant in sugar metabolism and carbon source sensing [23,24]. Through molecular genetics techniques, several studies have now discovered a correlation between hexokinase and spore yield, as well as toxin production of *Aspergillus fumigatus*; Fleck and Brock suggested that hexokinase is capable of maintaining high glucose phosphorylation activity, thereby controlling the synthesis of spore precursor materials such as trehalose, mannitol, and lipids [23]. Additionally, its influence on the interconversion between mannose-6-phosphate and fructose-6-phosphate impacts cell wall synthesis [23]. In recent years, studies have revealed that the absence of hexokinase weakens the toxicity of *Botrytis cinerea* on leaves, while exhibiting a significantly delayed nutritional growth and sharp reduction in conidial formation [25]. In studies of *Fusarium verticillioides*, hexokinase gene mutant showed significantly reduced toxicity towards maize and altered the biosynthesis of compounds such as trehalose and fumonisin B1 [26]. Therefore, the effect of hexokinase on morphogenesis and aflatoxin synthesis in *Aspergillus* spp. need to be explored.

In our previous study [27], it revealed that histone methyltransferases AshA and SetB regulates the H3K36me3 modification level of the chromatin fragment where hexokinase *afHxk1* gene is located, and given the grave economic, social, and health risks associated with *A. flavus*, as well as the pivotal role of hexokinase in morphogenesis and metabolic regulation, investigating the functional attributes of hexokinase in the various life activities of *A. flavus* holds immense significance. In this study, we explored in depth the biological functions of hexokinase AfHxk1 in *A. flavus* hyphal growth, sporulation, sclerotia development, aflatoxin production, and pathogenicity by construction of hexokinase gene mutants, key domain mutants, and crop and insect models, which provided valuable insights for potential novel control strategies.

## 2. Materials and Methods

### 2.1. Fungal Strains and Growth Conditions

The strains of *A. flavus* utilized in this research have been comprehensively recorded in Appendix A. The primers employed have also been meticulously tabulated in Appendix A. In order to facilitate the phenotypic analysis, the media below were used to inoculate the above fungal strains: potato dextrose agar (PDA, BD Difco™, NJ, USA), GMM (1.84 g/L ammonium tartrate, 0.52 g/L MgSO_4_·7H_2_O, 1.52 g/L KH_2_PO_4_, 10 g of glucose, 0.52 g/L KCl, and 1 mL trace elements), and YES media (150 g/L sucrose, 20 g/L yeast extract, 1 g/L MgSO_4_·7H_2_O) for morphogenesis analysis; potato dextrose broth (PDB, BD Difco™, NJ, USA) for mycotoxin analysis; and CM media (6 g/L yeast extract, 6 g/L peptone, 3 g/L sucrose) for sclerotia formation. Totally, 1.5% agar was added for solid media. Each fungal strain underwent a minimum of three plate cultures for biological replicate, with the experiment being repeated three times.

### 2.2. Bioinformatics Analysis

The AfHxk1 orthologs from ten species, including *Aspergillus flavus*, *Aspergillus oryzae*, *Aspergillus arachidicola*, *Aspergillus nomiae*, *Aspergillus niger*, *Aspergillus clavatus*, *Aspergillus nidulans*, *Schizosaccharomyces pombe*, *Arabidopsis thalian*, and *Mus musculus*, were obtained from the National Center for Biotechnology Information (NCBI, https://www.ncbi.nlm.nih.gov/, 12 December 2022) through a basic local alignment search tool algorithm. The neighbor-joining phylogenetic tree of the above proteins was constructed by MEGA 7.0. The domains in AfHxk1 were identified by SMART (https://smart.embl.de/, 12 December 2022), and were further visualized through DOG 2.0. To reveal the physicochemical properties of the AfHxk1, the ProtParam Tool (https://web.expasy.org/protparam/, 12 December 2022) was utilized to perform predictions regarding its isoelectric point, instability coefficient, and amino acid content.

### 2.3. Construction of Mutant Fungal Strains

Strain construction was performed on the principle of homologous recombination [28]. The *afHxk1* knock-out strain (∆*afHxk1*) method was carried out as shown in Appendix A. The upstream region (1627 bp), downstream region (1814 bp), and *pyrG* were amplified by PCR, using three pairs of primers (*afHxk1*-AF and *afHxk1*-AR, *afHxk1*-BF and *afHxk1*-BR, and *pyrG*-F and *pyrG*-R, respectively). Then, the above three fragments were overlapped together by further amplification with primer *afHxk1*-NF and *afHxk1*-NR. The fused PCR product was introduced into CA14 PTS via polyethylene glycol PEG-mediated transformation [29]. The selected transformants were tested by amplifying upstream homologous arm fragment (AP), downstream homologous arm (BP), and open reading frame (ORF) with three pairs of primers (*afHxk1*-AF and P801, P1020 and *afHxk1*-BR, and *afHxk1*-OF and *afHxk1*-OR, respectively). Those transformants with AP and BP and without ORF fragment were successfully constructed with ∆*afHxk1*.

The construction of the complementary strain (Com-*afHxk1*) was completed through two steps shown in Appendix A. First, the DNA fragment containing *afHxk1* and its upstream and downstream sequence was amplified with primers *afHxk1*-AF and *afHxk1*-BR, and was further transformed using the protoplasts of the Δ*afHxk1* strain. 5-fluoroorotic acid (5-FOA, 200 mg/mL) was added into the media to screen for the correct fungal strain (C1-*afHxk1*). Then, the upstream region (1699 bp) and downstream region (1628 bp) in the end of *afHxk1* were amplified by PCR, using two pairs of primers (Com-*afHxk1*-AF and Com-*afHxk1*-AR; Com-*afHxk1*-BF and Com-*afHxk1*-BR). The three fragments (the above two fragments and *pyrG* fragment amplified with primers *pyrG*-F and *pyrG*-R) were overlapped together by fusion PCR with primer Com-*afHxk1*-NF and Com-*afHxk1*-NR. The fusion product was used to transform the protoplasts of C1-*afHxk1*. Finally, the constructed complementary strain (Com-*afHxk1*) was further tested by diagnostic PCR following the method described above. The domain deletion fungal strains *afHxk1*^ΔD1^ and *afHxk1*^ΔD2^ were constructed and verified using the same method as for ∆*afHxk1*.

### 2.4. Real-Time Quantitative Reverse Transcription PCR

For the RT-qPCR analysis, the experimental procedure was conducted following the previously described method [30]. The mycelia of the tested fungal strains were collected from their respective culture media, which were subjected to all phenotypic analysis at a temperature of 29 °C for a duration of 2–3 d. Then, they were ground immediately in liquid nitrogen, and stored at −80 °C for subsequent RNA extraction. Total RNA extraction was performed using TRIzol reagent (Vazyme Biotech, Nanjing, China). First-strand cDNA synthesis was performed using HiScript III Enzyme Mix (Vazyme Biotech, Nanjing, China) according to the manufacturer’s instruction. The RT-qPCR analysis was performed using the Quantstudio 1 plus PCR system (Applied Biosystems, MA, USA). The primer sequences used in this study for RT-qPCR were listed in Appendix A. The expression level of β-tubulin gene was used as an internal reference. The relative expression levels of the target gene were calculated through the formula: 2^−ΔΔCt^.

### 2.5. Phenotypic and Carbon Source Analysis

According to the method used in the previous study, phenotypic analysis of the mutant fungal strains was performed [31]. The WT, Δ*afHxk1*, *afHxk1*^ΔD1^, *afHxk1*^ΔD2^, and Com-*afHxk1* strains were incubated in the dark at 37 °C on PDA, YES, and GMM culture media. After 5 d, the diameter of each colony was measured. Conidia of each sample were collected and washed with conidia collecting solution (0.05% Tween-20 and 7% DMSO). The collected conidia were counted by a hemocytometer under an optical microscope (Leica, Heerbrugg, Germany). For sclerotium analysis, CM medium was prepared to induce sclerotia formation, and the cultures were incubated with 10^4^ spore suspension in the dark at 37 °C for 7 d. Sclerotia were counted by taking equal-sized areas of small holes (1 cm^2^) along the radius. Each group consisted of a minimum of three replicates.

Previous studies have indicated that fungi exhibit a unique hexose kinase-mediated transport mechanism in the process of glucose metabolism [32]. Therefore, to analyze the regulatory role of AfHxk1 on fungal carbon source utilization capacity, GMM medium with different carbon sources (GMM-based carbon source media) were employed to inoculate the above fungal strains, in which the glucose of GMM was replaced with various alternative carbon sources (including fructose, mannose, xylose, sucrose, lactose, and starch). All fungal strains were inoculated onto the GMM-based carbon source media and cultured at 37 °C for 4 d. The colony diameter was finally measured, and photographic records were taken. All experiments were performed in triplicate.

### 2.6. Aflatoxin Analysis

To investigate the regulatory role of AfHxk1 on the production of AFs in *A. flavus*, the previous procedure was employed [29]. A spore suspension diluted to a concentration of 10^6^ spores/mL (10 μL) was added to 10 mL of PDB medium and cultured at 29 °C for 7 d. Subsequently, 4 mL of the culture was collected and mixed with 4 mL of dichloromethane for AFs extraction. After shaking on a rotary shaker at 29 °C for 30 min, 2 mL of the mixture was transferred to a fume hood for air drying. 

After subsequent air drying, the extract was redissolved with 5 mL of methylene chloride. Then, 5 mL of methanol solution (methanol:water, 55:45) was added for secondary extraction. The lower layer was collected and air dried and, finally, dissolved in 100 μL of dichloromethane. Then, the dissolved AFs sample (10 μL) was analyzed using silica gel plate (Qingdao Ocean New Material Technology, Qingdao, China) for thin-layer chromatography (TLC), and AFB_1_ standard solution (Sigma-Aldrich, Darmstadt, Germany) was used as the control. The mobile phase used for chromatography was the mixture of dichloromethane and acetone (9:1 *v*/*v*). After about half an hour, when the liquid phase slowly rose to the top of the TLC plate, the chromatography was stopped, and the TLC plate was quickly dried to eliminate any remaining solvent. Finally, the chromatographic results were documented using a UV gel imaging system (Beijing Oriental Science & Technology Development Ltd., Beijing, China).

### 2.7. Kernel Colonization Assays

To investigate the role of AfHxk1 in the colonization of *A. flavus* on crops, peanut and maize kernels were employed as the hosts for infection [33]. The kernels were surface-sterilized with 0.05% sodium hypochlorite, rinsed thoroughly with alcohol and sterile water, and then inoculated with spore suspensions of WT, Δ*afHxk1*, *afHxk1*^ΔD1^, *afHxk1*^ΔD2^, and Com-*afHxk1*. The infected kernels were incubated at 29 °C for 6 d. Then, the infected host kernels were collected in a centrifuge tube, and 10 mL of ddH_2_O was added and vortexed. The spore suspension (1 mL) was transferred, and the number of spores was counted using a hemocytometer. An additional 10 mL of dichloromethane was added into the host kernels contained in the centrifuge tube to extract AFs, and the steps for aflatoxin extraction were repeated following the above method introduced in this study. 

### 2.8. The Insect Infection Model

To explore the influence of AfHxk1 on animal infectivity by *A. flavus*, *Galleria mellonella* larvae were chosen as the host in the infection experiments. Firstly, fresh spore suspensions were diluted to 10^7^/mL and injected into each experimental sample (5 μL), while the control group (CK) was injected with an equal volume of saline. The injected larvae were kept in Petri dishes which were punctured and covered with moistened filter paper to provide appropriate ventilation and humidity. Subsequently, the Petri dishes were placed in a 37 °C incubator, with water added every 24 h to maintain humidity, and the mortality were recorded. After that, deceased larva samples were collected and placed in a 29 °C incubator for 7 d. Two samples from each replicate group were transferred to 50 mL centrifuge tubes containing ddH_2_O (10 mL), followed by vortex. Then, 1 mL of spore suspension was allotted for spore number assessing under a microscope with a hemocytometer. Subsequently, 10 mL dichloromethane was added to extract AFB_1_, and the steps for aflatoxin extraction following the method introduced in this study were performed. Each Petri dish contained ten larvae, and the experiment was repeated three times.

### 2.9. Stress Assays

To investigate the role of AfHxk1 in the environmental stress response of *A. flavus*, WT, Δ*afHxk1*, *afHxk1*^ΔD1^, *afHxk1*^ΔD2^, and Com-*afHxk1* strains were inoculated onto PDA agar supplemented with a series of concentrations of stress-inducing agents (including high osmotic stress media (1.0 M, 1.5 M, and 2.0 M KCl), oxidative stress agent (0.5 mM MSB, Menadione sodium bisulfite), cell membrane stress agents (0.01%, 0.02%, and 0.03% SDS, Sodium dodecyl sulfate), DNA-damaging agent (0.02% MMS, Methyl methanesulfonate), and cell wall stress agents (200 µg/mL and 300 µg/mL CFW, Calcofluor white, and 200 µg/mL and 300 µg/mL CR, Congo red). The cultures were incubated in the dark at 37 °C for 5 d. To analyze the involvement of AfHxk1 in the stress response of *A. flavus*, the relative inhibition rate was calculated using the formula: (colony diameter without inhibitor − colony diameter with inhibitor)/colony diameter without inhibitor. The experiment was repeated three times.

### 2.10. Statistical Analysis

In this study, all data were presented as means ± SD (standard deviation). The groups were compared for statistical significance using ANOVA and LSD (least significant difference) tests. Statistical analysis was performed using GraphPad Prism 5 software (La Jolla, CA, USA). Each experiment included triplicates for each sample, and the final results were confirmed through repetition three times. *p*-value of less than 0.05 was considered statistically significant.

## 3. Result

### 3.1. The Bioinformatic Analysis of AfHxk1 and the Construction of Fungal Mutants

To reveal the potential biofunction of AfHxk1 in *A. flavus*, the gene encoding AfHxk1 was screened and downloaded from NCBI (http://www.ncbi.nlm.nih.gov, 12 December 2022), and it was revealed that the gene is located on the first chromosome of *A. flavus*, with a coding protein sequence designation of XP_041141734.1. The orthologs of AfHxk1 from nine other species (*A. oryzae, A. arachidicola, A. nomiae, A. niger, A. clavatus, A. nidulans, Schizosaccharomyces pombe, Arabidopsis thaliana*, and *Mus musculus*) were obtained using the basic local alignment search tool (BLAST), and the evolutionary relationship of these homologs was constructed by MEGA 7.0 (Figure 1A), which showed that Hxk1 from *Aspergillus* spp. are grouped in one clade. The domains in Hxk1 were further identified on SMART and visualized by DOG 2.0, which revealed that all orthlogs contain two domains: Hexokinase_1 (Domain 1) and Hexokinase_2 (Domain 2), whereas *M. musculus* even possesses two pairs of the above domains. (Figure 1B). The value of the grand average of hydropathicity (GRAVY) was used to characterize protein hydrophobicity; analysis through ProtScale showed that the GRAVY value of AfHxk1 is −0.045, reflecting that it is a hydrophilic protein (Figure 1C). The fungal mutants were constructed following the method presented in Appendix A. The PCR results showed that AP and BP, but not ORF could be amplified from the genome of ∆*afHxk1*, and AP, BP, and ORF could be amplified from Com-*afHxk1* (Figure 1D). These results verified that the gene knock-out strain (∆*afHxk1*) and the complementary strain (Com-*afHxk1*) have been successful constructed. Further RT-qPCR demonstrated that no *afHxk1* was detected expressing in ∆*afHxk1*, while its expressing level recovered in Com-*afHxk1* compared to the wild type (WT) (Figure 1E). The sequencing of *afHxk1*^ΔD1^ and *afHxk1*^ΔD2^ showed that Domain 1 and Domain 2 have been knocked out, respectively (Appendix A).

### 3.2. AfHxk1 Promotes Fungal Capacity to Utilize Various Carbohydrates

To investigate the impact of AfHxk1 on the utilization of various carbon sources by *A. flavus*, the glucose in the GMM medium was replaced by a range of carbon sources, including fructose, mannose, xylose, sucrose, lactose, and starch, as mentioned previously. The fungal strains, WT, Δ*afHxk1*, *afHxk1*^ΔD1^, and *afHxk1*^ΔD2^, were inoculated on these MM (GMM with glucose)-based carbon source media for 4 d, then, their colonial diameters were recorded (Appendix A). The results revealed that the growth ability of Δ*afHxk1* and *afHxk1*^ΔD2^ was significantly inhibited on all types of carbon sources compared to WT, *afHxk1*^ΔD1^, and Com-afHxk1 strains. These findings suggested that AfHxk1 plays a crucial positive role in the utilization of a serial of carbon sources by *A. flavus*, in which Domain 2 is its key functional element, as *afHxk1*^ΔD2^ showed similar phenotype with that of Δ*afHxk1*.

### 3.3. AfHxk1 Is Deeply Involved in the Morphogenesis of A. flavus

To explore the role of AfHxk1 in the morphogenesis of *A. flavus*, WT, Δ*afHxk1*, *afHxk1*^ΔD1^, *afHxk1*^ΔD2^, and Com-*afHxk1* strains were inoculated on PDA, GMM, and YES media and cultured in the dark at 37 °C for 4 d. The results showed that the colony growth of Δ*afHxk1* and *afHxk1*^ΔD2^ were significantly slower than those of WT, *afHxk1*^ΔD1^, and Com-*afHxk1*, which is consistent with the trend in spore production on the above three different culture media (*p* < 0.001, *p* < 0.01) (Figure 2A–C). The conidiophore of the above fungal strains was further observed as shown in Figure 2D, which reflected that lower density of conidiophore were formed in both Δ*afHxk1* and *afHxk1*^ΔD2^ compared to WT and Com-*afHxk1* strains. Further RT-qPCR reflected that the absence of AfHxk1 reduced the production of conidia through downregulating the transcriptional levels of conidiation-related transcriptional factors AbaA and BrlA (*p* < 0.001) (Figure 2E). To explore the role of AfHxk1 in the sclerotium formation, the above *A. flavus* strains were inoculated on CM media and cultured in the dark at 37 °C for 7 d. The inoculation results showed that no sclerotium could be found from both Δ*afHxk1* and *afHxk1*^ΔD2^, while sclerotia were formed normally in other fungal strains (Figure 2F,H). RT-qPCR analysis suggested that the absence of AfHxk1 suppressed the production of sclerotia through sclerotium-related transcriptional factor NsdD and SclR (Figure 2G). No evidence was found in this study that Domain 1 is involved in the morphogenesis of *A. flavus* (Appendix A). These results indicated that AfHxk1 is essential for sclerotium formation, and its Domain 2 is the main element to fulfill its function in sclerotium formation.

### 3.4. AfHxk1 Negatively Regulates Aflatoxin Biosynthesis

To investigate the impact of AfHxk1 on AFB_1_ synthesis in *A. flavus*, the WT, Δ*afHxk1*, *afHxk1*^ΔD1^, *afHxk1*^ΔD2^, and Com-*afHxk1* strains were cultured in PDB media at 29 °C for 7 d. Then, the AFB_1_ production was exacted and analyzed by TLC, and the results revealed that, compared to the WT and Com-*afHxk1*, both Δ*afHxk1* and *afHxk1*^ΔD2^ mutants exhibited a significant rising trend in AFB_1_ production (Figure 3A,B), while it showed no significant difference between *afHxk1*^ΔD1^ and WT strains (Appendix A). Further RT-qPCR analysis showed that, when AfHxk1 was absent, the transcriptional levels of *aflC*, *aflD*, *aflO*, *aflR*, and *aflS* were significantly upregulated (*p* < 0.001) (Figure 3B). The above results suggested that AfHxk1, mainly via its Domain 2, significantly downregulates the AFB_1_ biosynthesis in *A. flavus* through classic aflatoxin gene clusters.

### 3.5. AfHxk1 Is Involved in the Colonization of A. flavus on Crop Kernels

In order to explore the effect of AfHxk1 on the colonization ability of *A. flavus* on crops, the kernels of peanut and maize were selected as the hosts in this study. Selected maize and peanut kernels were firstly sterilized, and then immersed in spore suspension (10^4^/mL). The kernels were cultivated in the dark at 29 °C for 5 d. The final results showed that, compared with WT and Com-*afHxk1*, the conidiation capacity of Δ*afHxk1* and *afHxk1*^ΔD2^ on the infected host kernels was significantly reduced (*p* < 0.001) (Figure 4A,B). The production of AFB_1_ in the infected peanut and maize kernels was further extracted with dichloromethane and analysis with TLC, respectively, and it was found that, as with the PDB media, the AFB_1_ production increased significantly in the host kernels infected by both Δ*afHxk1* and *afHxk1*^ΔD2^, compared to those infected by WT and Com-*afHxk1* strains (Figure 4C). The results also showed that the conidiation capacity and AFB_1_ biosynthesis level of *afHxk1*^ΔD1^ on the crop kernels were similar to those of WT and Com-*afHxk1* strains (Figure 4). The above results demonstrated that on the crop kernels, the deletion of AfHxk1 and its Domain 2 significantly decreased the sporulation capacity of *A. flavus*, but increased its AFB_1_ production. These also inferred the important role of Domain 2 in AfHxk1 in the colonization of *A. flavus* on crop kernels. In view of the findings that Domain 1 does not significantly take part in the fungal morphogenesis, AFB_1_ biosynthesis, and the pathogenesis of *A. flavus*, the following study mainly focuses on the AfHxk1 and its Domain 2 (i.e., focusing on Δ*afHxk1* and *afHxk1*^ΔD2^ mutants).

### 3.6. AfHxk1 Is Involved in Fungal Pathogenicity to Animal

To test the influence of AfHxk1 on the pathogenicity of *A. flavus* to animals, *Galleria mellonella* larvae were chosen as the animal host. 5 μL spore suspension (10^7^/mL) of the fungal strains (including WT, Δ*afHxk1*, and *afHxk1*^ΔD2^) was injected into the larvae. After 120 h of cultivation and observation, the larvae injected with Δ*afHxk1* and *afHxk1*^ΔD2^ showed significantly (at 48 h, 72 h, 96 h, and 120 h) lower survival rates compared to those injected with the WT strain (Figure 5A,B). The dead larvae were transferred and incubated under 29 °C for 7 d. Then, the conidia were counted with a hemocytometer, and the results showed that the absence of AfHxk1 significantly depressed the yield of conidiation (Figure 5C). Additionally, AFB_1_ extracted from dead larvae revealed a significant increase in the AFB_1_ synthesis ability in Δ*afHxk1* and *afHxk1*^ΔD2^, further corroborating the previous inference (Figure 5D). The above results indicated that AfHxk1 plays a crucial promotional role in the virulence of *A. flavus* towards infected insects, but significantly decreases fungal AFB_1_ biosynthesis capacity, and in the process, Domain 2 is its key functional element.

### 3.7. AfHxk1 Positively Regulates Fungal Spore Germination

To explore the role of AfHxk1 in the pathogenicity of *A. flavus*, the germination rates of WT, Δ*afHxk1*, *afHxk1*^ΔD2^, and Com-*afHxk1* strains were monitored. After inoculation on PDA media for 0 h, 3 h, 6 h, and 9 h, the germination rates of spores from mutant strains Δ*afHxk1* and *afHxk1*^ΔD2^ were assessed and compared to those of the WT and Com-*afHxk1* strains. The results showed that the germination rates of Δ*afHxk1* and *afHxk1*^ΔD2^ strains were significantly lower than those of the WT, but recovered in the Com-*afHxk1* strain at the 6th h (*p* < 0.01), indicating that the absence of AfHxk1 and its Domain 2 significantly inhibited fungal spore germination (Figure 6A,B). At the 9th h, the germination rate of both Δ*afHxk1* and *afHxk1*^ΔD2^ still remained significantly lower than those of the WT and control and Com-*afHxk1* strains (*p* < 0.001). To explore if the different germination rate resulted from the effect of different AFB_1_ yield produced by themselves, the spores of the above fungal strains were incubated under 0.04, 0.06, and 0.08 μg/mL AFB_1_ for 9 h, respectively, and the results showed that the concentration of AFB_1_ does not obviously affect the spore germination rate of all the above fungal strains (Appendix A). These findings suggested that AfHxk1 plays a crucial role in spore germination, and its Domain 2 is its key domain in the regulatory process.

### 3.8. AfHxk1 Plays Important Role in Fungal Susceptibility Responding to Stresses

To further explore the role of AfHxk1 in fungal pathogenicity, the biofunction of AfHxk1 in response of *A. flavus* to different environmental stresses was analyzed. The inhibitors, including KCl (mediated osmotic stress), SDS (inducing cell membrane stress), CFW and CR (inducing cell wall stress), MSB (mediated oxidative stress), and MMS (mediated DNA damage stress), were added into PDA media and the fungal strains (WT, Δ*afHxk1*, *afHxk1*^ΔD2^, and Com-*afHxk1*) were cultivated on the above media in the dark at 37 °C for 4 d. The results showed that the cell wall stressors CFW (200 and 300 μg/mL) and CR (200 and 300 μg/mL) significantly inhibited the hyphae growth of Δ*afHxk1* and *afHxk1*^ΔD2^, compared to that of WT. And, when *afHxk1* was reintroduced (i.e., Com-*afHxk1*)*,* the inhibition rate dropped back to that of the WT strain (Figure 7A–C). Gene *chsB* is crucial for hyphal growth, while *chsA* and *chsC* are involved in maintaining hyphal wall integrity [34]. Chitin transglycosylase is encoded by the gene *utr2*, and it is involved in cell wall assembly and regeneration [35]. Gene *mnpA* plays an important role in cell wall integrity and developmental patterns [36]. Further RT-qPCR analysis suggested that AfHxk1 might be responding to the CFW-mediated cell wall stress by ChsA- and ChsB-related signaling pathways, and responding to the CR-mediated cell wall stress by ChsB-, ChsC-, Utr2-, and MnpA-related signaling pathways (Figure 7D,E). To the KCl-mediated osmotic stress, the inhibition rates of KCl to Δ*afHxk1* and *afHxk1*^ΔD2^ strains were significantly higher than those of WT and Com-*afHxk1* at all concentration levels (Figure 7F,G). Gene *tcsB* functions as an osmotic sensor histidine kinase [37]. Gene *sln1* and *skn7* are involved in the SLN1-YPD1-SKN7 system that controls gene expression during osmotic stress [38]. RT-qPCR analysis showed that the expression levels of *tcsB*, *skn7*, and *sln1* in Δ*afHxk1* and *afHxk1*^ΔD2^ strains were significantly lower than those in the WT strain, suggesting that AfHxk1 responds to the KCl-mediated osmotic stress by TcsB-, Skn7-, and Sln1-related signaling pathways (Figure 7H). Under higher cell membrane stress mediated by 0.02% and 0.03% SDS, the inhibition rate to Δ*afHxk1* and *afHxk1*^ΔD2^ strains was significantly higher compared to WT and Com-*afHxk1* strains (Figure 7I,J). Rho1 is an important protein in the CWI pathway, regulating cellular processes [39]. SfaD is a negative regulator that impacts cellular resistance against SDS, while FlbA mediates the synthesis of glycoproteins on cell membranes [26,40,41]. Subsequent RT-qPCR analysis suggested that the absence of AfHxk1 upregulated FlbA- and Rho1-mediated signaling pathways and downregulated the SfaD-related signaling pathways to counteract or as a compensatory response to the SDS-induced restriction (Figure 7K).

The impact of AfHxk1 on the response to oxidative stress and DNA damage were also analyzed, and the results showed that AfHxk1 is significantly involved in the MSB-mediated oxidative stress and MMS-induced DNA damage stress (Appendix A). The above stress-related tests collectively suggested that the widespread participation of AfHxk1 in response to environmental stresses is an important pathway for regulation of the pathogenicity of *A. flavus*.

## 4. Discussion

Hexokinases are key enzymes for the metabolism of glucose and fructose, and are involved in glucose signaling from fungi to plants and animals [25]. Our previous study revealed that hexokinase AfHxk1 is involved in the regulation of fungal morphogenesis, mycotoxin synthesis, and pathogenicity mediated by Set2 family histone methyltransferase [27]. In view of the hazard of *A. flavus* to agriculture and human health, it is urgent to discover the role of the hexokinase AfHxk1 in the morphology, pathogenicity, and mycotoxin production in this notorious opportunistic fungal pathogen. In yeast, *hxk1* is an important gene that plays multiple roles in energy metabolism, gene transcription and expression regulation, glucose signaling, and filamentous growth [12,42,43,44], while in *Fusarium graminearum*, it has been found to play a crucial role in carbon catabolism, sporulation, and pathogenesis [26]. Additionally, mammals possess four important isozymes of hexokinase, which exhibit subcellular localization, kinetic properties towards different substrates and conditions, and physiological functions [21]. By the construction of phylogenetic tree and domain identification, it is revealed that AfHxk1 orthologs from *Aspergillus* spp. are relatively conserved (Figure 1A,B), which reflected that AfHxk1 might play a very important role in the biologic function of *A. flavus.* However, there have been no reports on the regulatory role of AfHxk1 in the phenotype, secondary metabolism, and pathogenicity of *A. flavus*. Our study discovered the biofunction of AfHxk1 in *A. flavus* in further enriching the epigenetic regulatory mechanisms mediated by the Set2 family histone methyltransferase.

### 4.1. AfHxk1 Enhances Fungal Colonial Growth and Conidiation

Previous studies have demonstrated that Hxk1 is involved in the yeast-to-hyphal transition in *Candida albicans* and negatively regulates spore formation in *F. verticillioides* [44,45]. To investigate the role of AfHxk1 in the growth and sporulation of *A. flavus*, related *A. flavus* strains were inoculated on a series of media (including PDA, GMM, and YES), and it revealed that the deletion of AfHxk1 and its Domain 2 resulted in a significant decrease in conidial density and slower colony growth at 37 °C, reflecting that, mainly through Domain 2, AfHxk1 upregulates hyphal growth and asexual reproduction of this infamous filamentous pathogenic fungus (Figure 2A–D). In the experiment with MM-based carbon source media, we also observed a significant downregulation of the carbon source utilization ability of Δ*afHxk1* and *afHxk1*^ΔD2^ (Appendix A). Previous studies have shown that Hxk1 knockout *C. albicans* strains are unable to grow on fructose, reflecting that there possibly are alternative metabolic pathways in *A. flavus* for fructose metabolism compared to *C. albicans* [46]. In the crop kernel invading experiments, similar results were found in the aspect of conidial formation, with significantly lower conidial numbers in Δ*afHxk1* and *afHxk1*^ΔD2^ strains compared to WT and Com-*afHxk1*. Further RT-qPCR analysis revealed that AfHxk1 upregulates the conidiation of *A. flavus* through increasing the expression levels of *brlA* and *abaA* (Figure 2E). BrlA and AbaA are important canonical regulatory factors of asexual reproduction in the *Aspergillus* spp., activating downstream conidium-specific gene expression in a hierarchical manner during conidiophore development and conidium formation and maturation [47]. Therefore, the related results of this study proposed that AfHxk1, mainly depending on its Domain 2, enhances hyphal growth by promoting fungal capacity to utilize various carbohydrates, and promotes asexual reproduction by upregulating sporulation central regulatory pathways (BrlA→AbaA→WetA), finally realizing the control of *A. flavus* hyphal growth and conidia synthesis.

### 4.2. AfHxk1 Is Indispensable for the Formation of Sclerotia

Sclerotia are compact masses of hyphae that certain fungi produce as a survival structure under unfavorable environmental conditions [48]. In many *Aspergillus* species, sclerotia are a substrate for the formation of sexual structures [48]. This study found that the ability of sclerotium formation was completely lost in Δ*afHxk1* and *afHxk1*^ΔD2^ strains, accompanied by the downregulation of the expression levels of *nsdD* and *sclR*. The NsdD*,* a GATA-type transcription factor, is a key regulator in sclerotia development in *A. nidulans* [48]. And the *sclR* gene also plays a crucial role in sclerotia formation of *A. flavus* [49]. The results of this study suggested that AfHxk1 initiates and promotes the formation of sclerotia through activation the pathways of NsdD and SclR, and the process is mainly dependent on its Domain 2. Consequently, through promoting sexual reproduction, AfHxk1 may also promote genetic variation in *A. flavus*, enhancing environmental adaptability of this pathogenic fungus. In brief, AfHxk1 is obviously advantageous for the survival of *A. flavus* in unfavorable environments, in which its Domain 2 plays an indispensable role.

### 4.3. AfHxk1 Downregulates the Biosynthesis of AFB_1_

Aflatoxins are the most notorious mycotoxins, among which AFB_1_ is even more harmful. This study found that AfHxk1 significantly downregulates the biosynthesis of AFB_1_ in PDB, and on the crop and insect infection models (as shown in Figure 3, Figure 4 and Figure 5). Further RT-qPCR reflected that it downregulates AFB_1_ yield by aflatoxin biosynthesis gene cluster, including *aflR*, *aflS*, *aflC*, *aflD*, and *aflO* (Figure 3B). AflR is a transcription factor that plays a crucial role in regulating genes in the aflatoxin biosynthesis gene cluster [45]. AflR promotes the yield of AFB_1_ by activating genes such as *aflC*, *aflD*, and *aflO*, while inhibiting genes like *aflJ* and *aflM* [45]. Adjacent to *aflR*, the *aflS* gene is also essential for regulating aflatoxin biosynthesis through its interaction with *aflR* and disruption of this interaction may lead to a decrease in aflatoxin production [50]. *aflC* and *aflD*, which are also regulated by *aflS*, are the structural genes involving in AFB_1_ synthesis [45,51]. Gene *aflC* catalyzes the chemical reaction of acetate to norsolorinic acid [45,51]. While *aflD* encodes an enzyme that converts norsolorinic acid into the AFB_1_ precursor averantin. And AflO participates in the key pathway of catalyzing the conversion of DMST to ST in aflatoxin biosynthesis [51]. Therefore, it is speculated that AfHxk1, mainly through its Domain 2, regulates AFB_1_ biosynthesis by inhibiting the aflatoxin biosynthesis gene cluster in *A. flavus*. 

### 4.4. AfHxk1 Plays a Vital Role in Fungal Pathogenicity

*A. flavus* wildly invades various crops and randomly infects animals with lower immunity, so it is important to explore the role of AfHxk1 in fungal pathogenicity. The results of this study reflected that AfHxk1 plays a crucial role in the pathogenicity of *A. flavus* in both crop grains and insect larvae. As previously mentioned, the absence of AfHxk1 led to a significant reduction in conidial production in both crop and insect models (Figure 4A,B and Figure 5A,C). 

To reveal the underlying mechanism, the analysis of spore germination was carried out, and the results revealed that the absence of AfHxk1 significantly inhibited spore germination at the 6th and 9th h (Figure 6), which might be related to its weaker carbon source utilization rate (Appendix A). The fungal ability to infect the host is closely related to its ability to cope with various environmental stresses. To explore the role of AfHxk1 in the response of *A. flavus* to different environmental stresses, the inhibitors, including KCl, SDS, CFW and CR, MSB, and MMS are tested, and the results showed that AfHxk1 wildly participates in *A. flavus* response to environmental stresses, including osmotic stress, cell membrane stress, cell wall stress, oxidative stress, and DNA damage stress (Figure 7). Fungal cells can dynamically modify their cell wall architecture to adapt to stress and evade the host immune system [52]. The *chsB*, a chitin synthase gene, is crucial for normal hyphal growth, while *chsA* and *chsC* are necessary for hyphal wall integrity [34]. Gene *utr2* encodes chitin transglycosylase, an enzyme involved in cell wall assembly and regeneration [35]. Gene *mnpA* encodes a mannosylglycoprotein in *A. nidulans* that plays a role in cell wall integrity and developmental patterns [53]. The results of this study revealed that the absence of AfHxk1 was accompanied by a significant decrease in the expression of *chsA*, *chsB*, *chsC*, *utr2*, and *mnpA*, resulting in increased sensitivity of *A. flavus* to cell wall stressors CFW and CR (Figure 7D,E).

Under KCl-mediated osmotic stress, the sensitivity of AfHxk1 mutant strain increased compared to WT and Com-*afHxk1*, and RT-qPCR showed that the expression levels of *tcsB*, *skn7*, and *sln1* decreased in both Δ*afHxk1* and *afHxk1*^ΔD2^ (Figure 7H). TcsB is a homolog of the yeast osmosensor Sln1p and encodes a transmembrane hybrid-type histidine kinase that functions as an osmosensor histidine kinase [37]. Gene *sln1* encodes a histidine kinase and is part of the SLN1-YPD1-SKN7 two-component regulatory system, which controls gene expression in response to osmotic stress [38]. This study proposed that AfHxk1 contributes to the response of *A. flavus* to osmotic stress inhibitors by upregulating the expression levels of *tcsB* and *sln1*, and thereby affecting the expression of *skn7*, and Domain 2 is the key functional element in the process.

Under SDS-mediated cell membrane stress, the growth of Δ*afHxk1* is significantly inhibited compared to WT and Com-*afHxk1* (Figure 7I,J). RT-qPCR analysis suggested that, to compensate the growth restriction caused by the absence of AfHxk1-, FlbA-, and Rho1-mediated signaling pathways were upregulated, and the SfaD-activated pathway was downregulated (Figure 7K). SDS-mediated stress triggers a kinase cascade known as cell wall integrity (CWI) pathway, which responds to disturbances in the cell wall and cell membrane to maintain cell integrity [54]. As an important regulatory protein within the CWI pathway, the expression level of the G protein Rho1 is significantly upregulated under SDS-mediated stress [39]. FlbA and G proteins play a collaborative role in regulating fungal morphogenesis [40]. The results of this study suggested that the CWI pathway is enhanced in response to SDS stress in the Δ*afHxk1* strain to compensate for the restriction induced by the lack of AfHxk1. SfaD is a negative regulatory factor involved in cellular resistance to SDS [41]. The results of this study indicated that in the absence of AfHxk1, *A. flavus* might compensate or counteract SDS-mediated cell membrane stress by downregulation of the SfaD pathway.

The filamentous pathogenic fungus *A. flavus* heavily contaminates various agricultural products and invades immune deficient animals, producing the most toxic mycotoxin AFB_1_. This study revealed that AfHxk1 plays a key role in the development of AFB_1_ biosynthesis and virulence through serial related classical signaling pathways, carbon sources utilization ability, spore germination rate, and sensitivity response to environmental stresses. And the Domain 2 of AfHxk1 was identified to be its key functional element in implementing all of the above important biofunctions. This study reveals the important biological role of AfHxk1 in *Aspergillus* spp., and provides a novel potential target for the control of the contamination of *A. flavus* and its mycotoxin AFB_1_.

## Figures and Tables

**Figure 1 jof-09-01077-f001:**
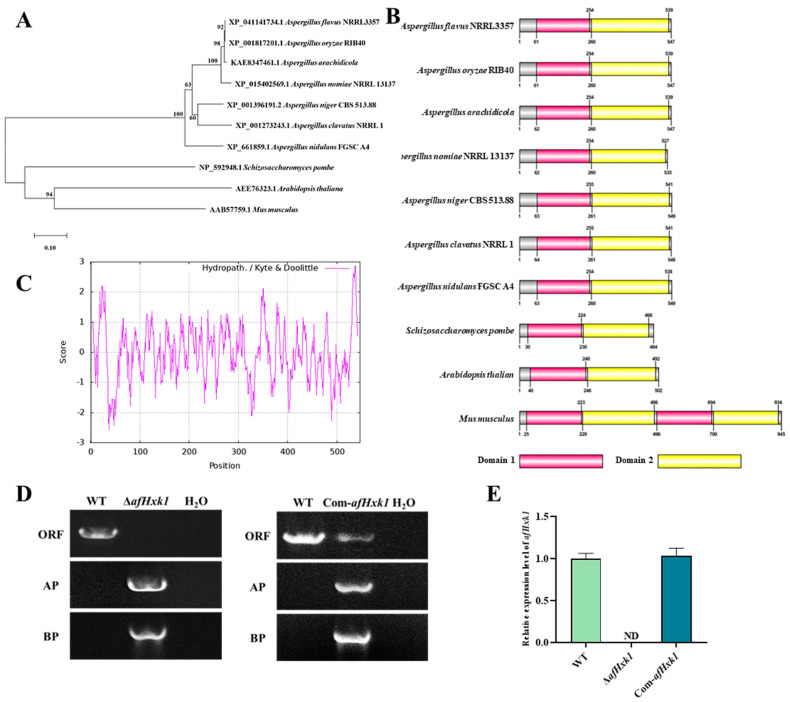
The bioinformatics analysis of AfHxk1 and the construction of *A. flavus* mutants. (**A**) Phylogenetic tree analysis of Hxk1 orthologs from ten species, including *A. flavus* (XP_041141734.1), *A. oryzae* (XP_001817201.1), *A. arachidicola* (KAE8347461.1), *A. nomiae* (XP_015402569.1), *A. niger* (XP_001396191.2), *A. clavatus* (XP_001273243.1), *A. nidulans* (XP_661859.1), *S. pombe* (NP_592948.1), *A. thaliana* (AEE76323.1), and *M. musculus* (AAB57759.1). (**B**) Domain identification of AfHxk1 from the above ten species, visualized by DOG 2.0. Domain 1: Hexokinase_1; Domain 2: Hexokinase_2. (**C**) Hydrophobic analysis of AfHxk1 showed that it was a hydrophilic protein. (**D**) The verification of the ∆*afHxk1* and Com-*afHxk1* strains by diagnostic PCR. In ∆*afHxk1*, *afHxk1* gene was replaced by *pyrG*, so AP and BP could be amplified from the genome of ∆*afHxk1*, but ORF could not. While ORF, AP, and BP all could be amplified from the genome of Com-*afHxk1*. (**E**) Relative expression levels of *afHxk1* in WT, ∆*afHxk1*, and Com-*afHxk1* strains. ND represents not detected.

**Figure 2 jof-09-01077-f002:**
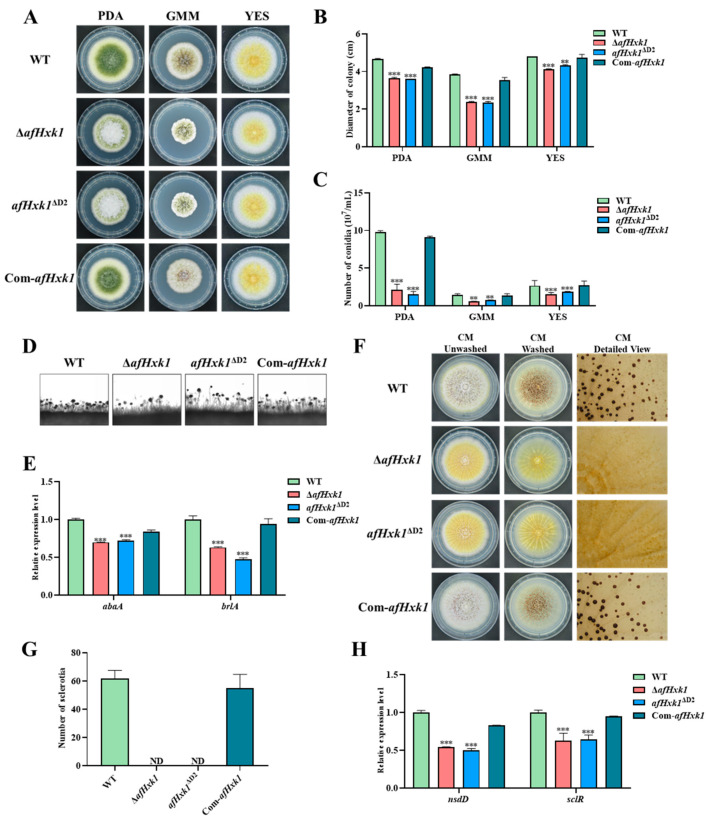
The role of AfHxk1 and Domian 2 played in the growth and development of *A. flavus*. (**A**) The growth of the fungal strains (including WT, Δ*afHxk1*, *afHxk1*^ΔD2^, and Com-*afHxk1*) on PDA, GMM, and YES at 37 °C in the dark for 4 d. (**B**) The colonial diameter of the above fungal strains on the three kinds of media. (**C**) The conidia number of the above fungal strains on the three different media. (**D**) The conidiophores of the above fungal strains on PDA. (**E**) Relative expression levels of *abaA* and *brlA*. (**F**) Sclerotium formation of the above fungal strains. The hyphae and conidia were washed away by 75% ethanol for detailed view. (**G**) Statistical analysis of the sclerotia produced by the above fungal strains. (**H**) Relative expression of *nsdD* and *sclR*. **, *** means significant difference *p* < 0.01, *p* < 0.001. ND represents not detected.

**Figure 3 jof-09-01077-f003:**
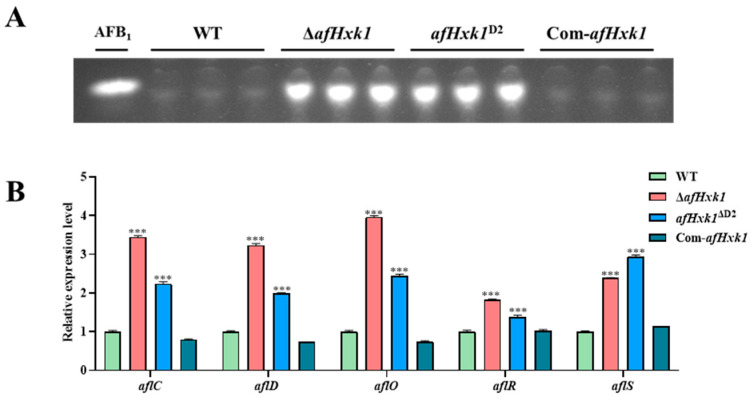
The effect of AfHxk1 on the aflatoxin synthesis from *A. flavus*. (**A**) The production of AFB_1_ produced by WT, Δ*afHxk1*, *afHxk1*^ΔD2^, and Com-*afHxk1* strains was detected by TLC. The aflatoxin B1 was exacted by dichloromethane. (**B**) Relative expression levels of *aflC*, *aflD*, *aflO*, *aflR*, and *aflS* (the aflatoxin-producing structural genes and regulatory genes). *** means significant difference *p* < 0.001.

**Figure 4 jof-09-01077-f004:**
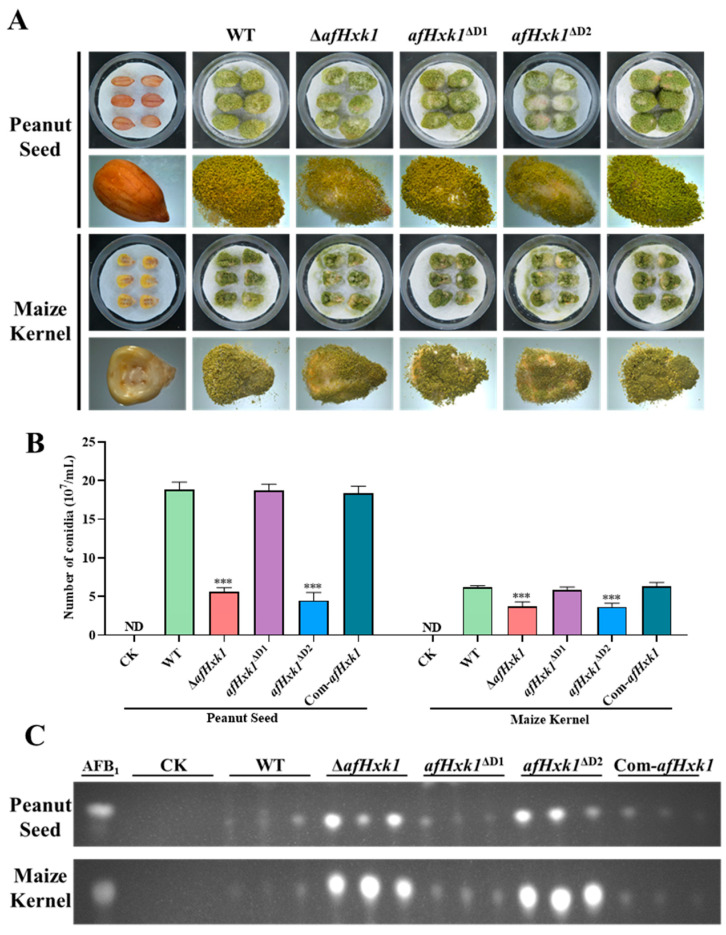
AfHxk1 is involved in the colonization capacity of *A. flavus* on crops. (**A**) The colonization of WT, Δ*afHxk1*, *afHxk1*^ΔD1^, *afHxk1*^ΔD2^, and Com-*afHxk1* strains on peanut seeds and maize kernels at 29 °C in the dark for 5 d. (**B**) The statistics on the conidia number from the above fungal strains on the peanut seeds and maize kernels. (**C**) The TLC analysis on the AFB_1_ yield produced by the above fungal strains on the kernels. *** means significant difference, *p* < 0.001. ND represents not detected.

**Figure 5 jof-09-01077-f005:**
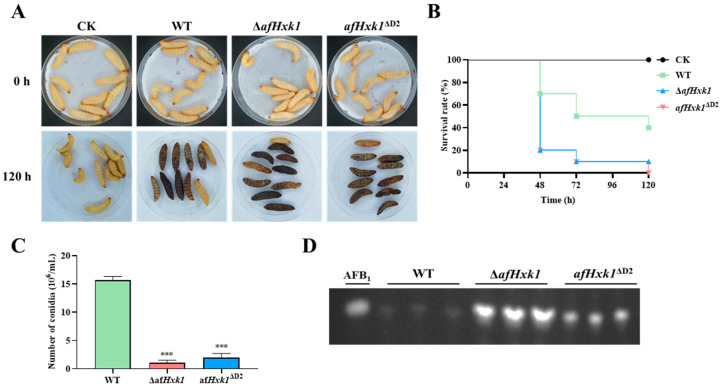
AfHxk1 is involved in the pathogenicity of *A. flavus* to *G. mellonella*. (**A**) The *G. mellonella* larvae infected with the spores of fungal strains: WT, Δ*afHxk1*, and *afHxk1*^ΔD2^. The larvae injected with saline were set as control. The dead larvae were collected and further cultured at 29 °C in the dark for 7 d. (**B**) The survival rate of the infectious larvae. (**C**) Counting of conidia of the aforementioned fungal strains grown on the dead larvae. (**D**) The TLC analysis of AFB_1_ extracted from the dead larvae. *** means significant difference *p* < 0.001.

**Figure 6 jof-09-01077-f006:**
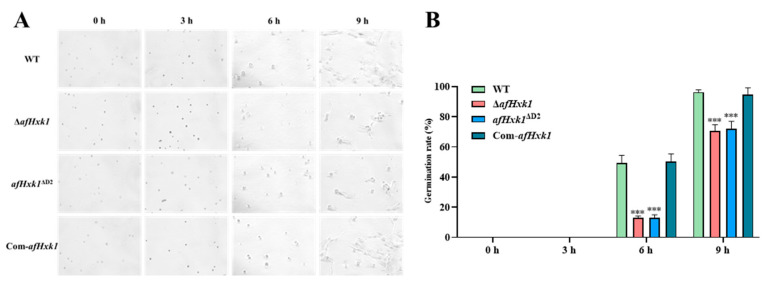
The effect of AfHxk1 on fungal spore germination. (**A**) The germination of fungal spores. The spores of WT, Δ*afHxk1*, *afHxk1*^ΔD2^, and Com-*afHxk1* were inoculated on PDA for 0 h, 3 h, 6 h, and 9 h, and the results were documented and observed with microscope. (**B**) Statistics of germination rates of the aforementioned spores at the above time points. *** means significant difference, *p* < 0.001.

**Figure 7 jof-09-01077-f007:**
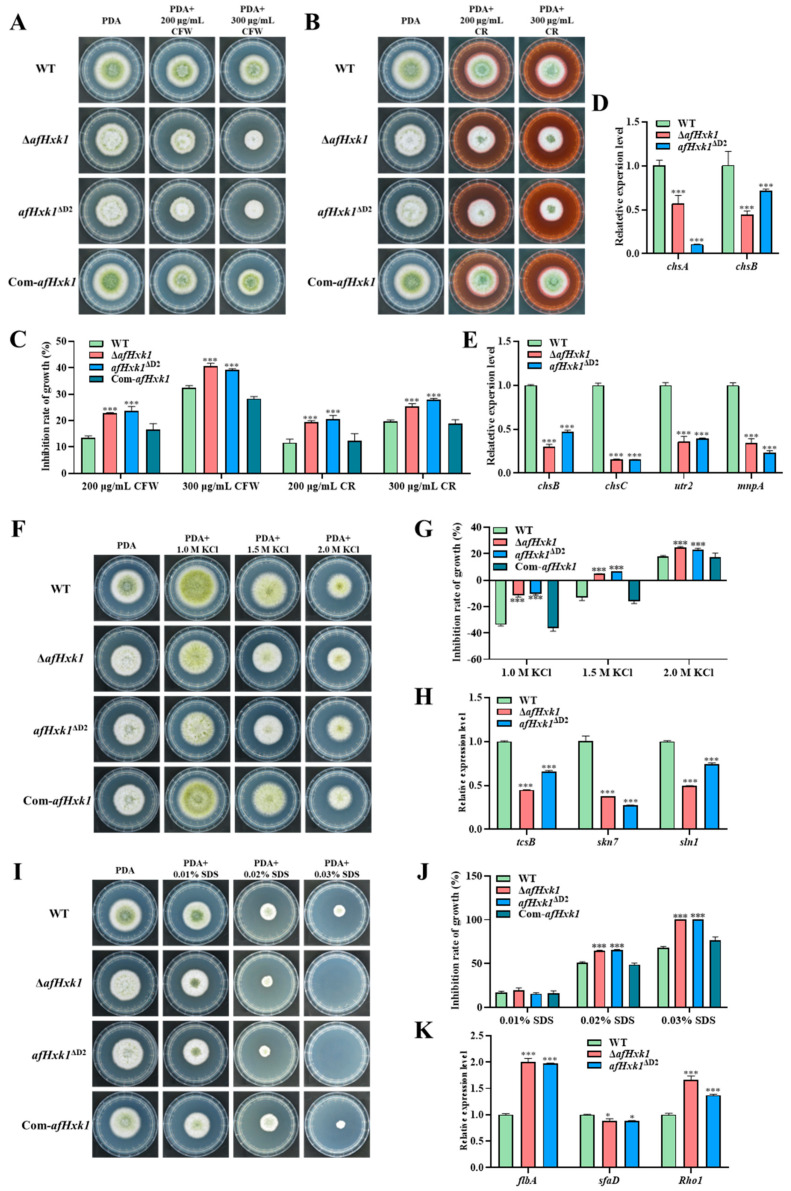
AfHxk1 wildly participates in *A. flavus* response to various environmental stresses. (**A**) The fungal strains under the cell wall stress mediated by CFW. The spores of WT, Δ*afHxk1*, *afHxk1*^ΔD2^, and Com-*afHxk1* were inoculated on PDA with 200 μg/mL and 300 μg/mL CFW for 4 d. (**B**) The above fungal strains under the cell wall stress mediated by CR. The fungal spores of the aforementioned strains were inoculated on PDA mediated with 200 μg/mL and 300 μg/mL CR. (**C**) The inhibition rate of fungal growth under cell wall stress mediated by CFW and CR. (**D**) Relative expression levels of *chsA* and *chsB* under cell wall stress mediated by CFW (200 μg/mL). (**E**) Relative expression levels of *chsB*, *chsC*, *utr2*, and *mnpA* under cell wall stress mediated by CR (200 μg/mL). (**F**) The fungal strains under the osmotic stress mediated by KCL. The spores of the above fungal strains were inoculated on PDA with 1.0 M, 1.5 M, and 2.0 M KCl. (**G**) The inhibition rate of fungal growth under osmotic stress mediated by KCl. (**H**) Relative expression levels of *tcsB*, *skn7*, and *sln1* under osmotic stress mediated by KCl (1.0 M). (**I**) The fungal strains under the cell membrane stress mediated by SDS. The spores of the above strains were inoculated on PDA with 0.01%, 0.02%, and 0.03% SDS. (**J**) The inhabitation rate of fungal growth under cell membrane stress mediated by SDS. (**K**) Relative expression levels of *flbA*, *sfaD,* and *Rho1* under cell membrane stress mediated by SDS (0.01%). *, *** means significant difference, *p* < 0.05, *p* < 0.001, respectively.

## Data Availability

The authors confirm that the data supporting the findings of this study are available within the article and its Appendix A.

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
