# Peer review of "Regulation of Fungal Morphogenesis and Pathogenicity of Aspergillus flavus by Hexokinase AfHxk1 through Its Domain Hexokinase_2"

_jof, 2023, doi:10.3390/jof9111077_

Round 1

Reviewer 1 Report

Comments and Suggestions for Authors

The manuscript by Huang et al. characterized the A. flavus Hxk1 gene by creating the deletion mutants and complementing the mutant to wild type phenotype through putting back the functional copy of gene or domain. The manuscript is well writing and data were presented in a logical fashion. The data are new and interesting. Therefore, the manuscript is suitable for publishing in JOF if the authors can address the following concerns satisfactorily:

11.     My main concern of this study is that the authors seem to lack some basic knowledge in working with A. flavus as well as some fundamental lab techniques. If the studies were conducted exactly as they said in the M and M, then their data are questionable. I just hope these are the results of some kind of blatant oversights. If yes, then a serious explanation is needed to convince me how these mistakes in the manuscript slit through so many authors and no one caught them. Here are a few examples: First, line 195-196: “Conidia of each sample were collected by washed off using 7% Tween-20 solution”. I have never seen any study that used such a high concentration of Tween-20 in resuspending A. flavus conidia in my over 25 years of career working with this fungus. The normal concentration I have seen is around 0.01 to 0.05%. Second, lines 231-232: The kernels were surface-sterilized with 0.05% sodium hypochlorite, rinsed thoroughly”. The duration of seed sterilization was not specified here. The concentration of sodium hypochlorite used for seed surface sterilization is less than 1/10 of the concentration that is normally used (about 0.8%). I have seen people use 1-5% for 1 min. 0.05% is too low to be effective. Third, the optimum temperature for culturing A. flavus is around 29-31 C. When grown under 37C, its aflatoxin production ability is severely reduced based on published study. The authors gave no justification in using 37C in a few of the experiments conducted in this study (lines 357, 557, etc). Fourth, line 433 to 434: “The dead larvae were transferred and inoculated under 29 for 7 d”. Why would you inoculate the dead larvae again? This does not make any sense. I just hope you mean “incubate” instead of “inoculate”.

22.   The authors seem to have a misunderstanding on what methods to remove aflatoxin are considered pre-harvest or post-harvest. All the examples listed in lines 60-61 are post-harvest. None was pre-harvest.

33.       For extracting aflatoxin from infected kernels (lines 229-140), besides the sodium hypochlorite concentration concern, the chemical called “dichloride” was used for extracting aflatoxin from kernels. There are two other concerns here: first dichloride is not a full name of a chemical and I do not understand why such an error occurred. Second, there was no mention of grounding the kernels in order to extract aflatoxin from inside kernel. This may not be necessary if you have documentation to show the amount of aflatoxin extracted from fungal mycelia that grew outside the kernels is proportional to the amount of toxin inside the kernels.

44.       Figure 6. How do you rule out the lower germination rate was not caused by high levels of aflatoxin being produced in these mutants???

55       The justification on examining the specific genes in the stress study need to be added in the method section or in the result section. A short version of what you have in the discussion section (lines 569-572, 585-588, 602-613, 635-640, 647-651, 659-665) should do.

66.      There are some editorial corrections that are indicated in the returned edited pdf file. Please incorporate these changes during your revision.

Comments on the Quality of English Language

The english is fine.

Author Response

11.My main concern of this study is that the authors seem to lack some basic knowledge in working with A. flavus as well as some fundamental lab techniques. If the studies were conducted exactly as they said in the M and M, then their data are questionable. I just hope these are the results of some kind of blatant oversights. If yes, then a serious explanation is needed to convince me how these mistakes in the manuscript slit through so many authors and no one caught them.

Here are a few examples:

 First, line 195-196: “Conidia of each sample were collected by washed off using 7% Tween-20 solution”. I have never seen any study that used such a high concentration of Tween-20 in resuspending A. flavus conidia in my over 25 years of career working with this fungusThe normal concentration I have seen is around 0.01 to 0.05%.

Answer 11: Thank you, we have a writing mistake. In the experiment, we mixed 0.5 mL Tween-20 into 1000 mL water, so the final concentration should be 0.05%. And the final conidia collecting solution are: 0.05% Tween-20 solution with 7% DMSO. It had been revised in the Line 196-197.

 Second, lines 231-232: The kernels were surface-sterilized with 0.05% sodium hypochlorite, rinsed thoroughly”. The duration of seed sterilization was not specified here. The concentration of sodium hypochlorite used for seed surface sterilization is less than 1/10 of the concentration that is normally used (about 0.8%). I have seen people use 1-5% for 1 min. 0.05% is too low to be effective.

Answer 12: Thank you, we have double checked it. To prepare the solution, 625 μL 8% sodium hypochlorite was added into 100 mL water, so the final concentration is 0.05%. And the duration of seed sterilization was 3 min. Under this condition, our CK group were perfectly sterilized as showed in Figure 4A. The reference paper (please refer to Page 6 of this paper) using the same lower concentration are provided below:

Kale SP, Milde L, Trapp MK, Frisvad JC, Keller NP, Bok JW. Requirement of LaeA for secondary metabolism and sclerotial production in Aspergillus flavus. Fungal Genet Biol. 2008 Oct;45(10):1422-9. doi: 10.1016/j.fgb.2008.06.009. Epub 2008 Jul 11. PMID: 18667168; PMCID: PMC2845523.

 Third, the optimum temperature for culturing A. flavus is around 29-31 C. When grown under 37C, its aflatoxin production ability is severely reduced based on published study. The authors gave no justification in using 37C in a few of the experiments conducted in this study (lines 357, 557, etc).

Answer 13: In our experiments, 29-30oC is used to observe aflatoxin biosynthesis, but this fungus grows/develops (including hypae growth, conidaiton, and sclerotium formaiton) faster under 37oC. So the higher temperature (37oC in this manuscript) was used to observe the development of A. flavus, and lower temperature (29oC in this manuscript) was used in the study of aflatoxin yield and invasion experiments. The reference paper using these temperature points are provided below: (please refer to Page 18 and 19 of the referring paper)

Tan C, Deng JL, Zhang F, Zhu Z, Yan LJ, Zhang MJ, Yuan J, Wang SH. CWI pathway participated in vegetative growth and pathogenicity through a downstream effector AflRlm1 in Aspergillus flavus. iScience. 2021 Sep 23;24(10):103159. doi: 10.1016/j.isci.2021.103159. PMID: 34693219; PMCID: PMC8517163.

Fourth, line 433 to 434: “The dead larvae were transferred and inoculated under 29℃ for 7 d”. Why would you inoculate the dead larvae again? This does not make any sense. I just hope you mean “incubate” instead of “inoculate”.

Answer 14: Thank you very much! it should be “incubate”, we have corrected it according to your kind advice in Line 428.

  1. The authors seem to have a misunderstanding on what methods to remove aflatoxin are considered pre-harvest or post-harvest. All the examples listed in lines 60-61 are post-harvest. None was pre-harvest.

Answer 2: Thank you, we have moved all the original example to post-harvest and added a reference for pre-harvest according to the paper showed below (please refer to Page 11 and 12 of this reference):

Pickova, D., Ostry, V., Toman, J., & Malir, F. (2021). Aflatoxins: History, significant milestones, recent data on their toxicity and ways to mitigation. Toxins, 13(6), 399.

  1.  For extracting aflatoxin from infected kernels (lines 229-140), besides the sodium hypochlorite concentration concern, the chemical called “dichloride” was used for extracting aflatoxin from kernels. There are two other concerns here: first dichloride is not a full name of a chemical and I do not understand why such an error occurred.

Answer 31: Thank you very much, it should be “dichloromethane”. The correction has been made in Line 243 and 260 based on your kind suggestion.

Second, there was no mention of grounding the kernels in order to extract aflatoxin from inside kernel. This may not be necessary if you have documentation to show the amount of aflatoxin extracted from fungal mycelia that grew outside the kernels is proportional to the amount of toxin inside the kernels.

Answer 32: Thank you for your valuable feedback on this problem. Regarding the extraction of toxins from infected seeds, there already are some published works direct extraction of these toxins from seeds without previously ground, one of these papers are showed below (please refer to Page 3 of this referring paper). In view of most aflatoxins are produced on the adjacent areas of kernel surface, it did, at least partly, reflect the change trend of AFs biosynthesis ability of these studied A. flavus strains. And we will further optimize our experiments in our following related experiments according to your nice advice.

Lan, H., Wu, L., Fan, K., ... & Wang, S. (2019). Set3 is required for asexual development, aflatoxin biosynthesis, and fungal virulence in Aspergillus flavus. Frontiers in Microbiology, 10, 530.

  1.      Figure 6. How do you rule out the lower germination rate was not caused by high levels of aflatoxin being produced in these mutants???

Answer 4: Thank you for this good question. As mentioned previously, we used 37°C to culture the fungal strains to observe various phenotypes at a higher growth speed, including germination. However, as we all known, such high temperature seriously inhibits the biosynthesis of AFB1. In fact, at this high temperature, AFB1 production is reduced by at least 70%-90% or even completely inhibited. According to the above analysis, it can be concluded that AFB1 has very little (if has) effect on spore germination. The reference paper are provided below (please refer to Page 10 of this reference):

Gallo, A., Solfrizzo, M., Epifani, F., Panzarini, G., & Perrone, G. (2016). Effect of temperature and water activity on gene expression and aflatoxin biosynthesis in Aspergillus flavus on almond medium. International journal of food microbiology, 217, 162-169.

We also wonder whether AFB1 could affect the germination of spores, so a experiments were designed according to your good advice, in which the spores of WT and gene deletion fungal strain were inoculated in a series of AFB1 concentration, and the final germination rate are calculated as showed in Figure S5 of this manuscript. The results confirmed that AFB1 has no obvious effect on spore germination.

55       The justification on examining the specific genes in the stress study need to be added in the method section or in the result section. A short version of what you have in the discussion section (lines 569-572, 585-588, 602-613, 635-640, 647-651, 659-665) should do.

Answer 5: Thank you, we have justified all the qPCR genes related to the stress study according to your kind advice.

  1.     There are some editorial corrections that are indicated in the returned edited pdf file. Please incorporate these changes during your revision.

Answer 6: Thank you very much! We’ve carefully checked and corrected our manuscript according the comments in the pdf file.

Question in the PDF file:

Line 156:Thank you, we have tried to put Fig S1 into the main text Fig.1, but this seems to cause the layout of the picture in each panel to be too small and crowded, and the clarity were obviously reduced. In view of the combination of these two figures would significantly decrease the readers' reading experience, authors tend not to combine these two figures after in-depth discussion.

Reviewer 2 Report

Comments and Suggestions for Authors

The manuscript presents and analyzes the impact of genomic modification of the Aspergillus flavus exokinase AfHxk1, demonstrating that this genotype has a profound effect on fungal growth, carbon source utilization, conidial germination, stress sensitivity and aflatoxin B1 production as well as pathogenicity to maize grain and Galleria mellonella larvae. The manuscript presents an in-depth study of the role of the exokinase AfHxk1 and thus envisions the possibility of manipulating Aspergillus flavus to avoid the deleterious effect of the fungus on human and animal health as well as agriculture. However, the manuscript has several deficiencies that should be corrected before publication.

The title of the manuscript (AfHxk1 regulates fungal morphogenesis and pathogenicity of Aspergillus flavus through its domain Hexokinase_2) seems difficult to understand for the lay audience, partly because it starts with a name independent (AfHxk1) from the context of the study. Therefore, it is suggested to reconsider the construction of a simpler title (For example: Regulation of fungal morphogenesis and pathogenicity of Aspergillus flavus through its domain of the Hexokinase AfHxk1).

The abstract does not seem to clearly represent the breadth and depth of the study, as it contains a broad introduction and interpretation, while the methods and results seem to occupy a smaller proportion than expected. It is suggested that the methods and results be presented more fully and schematically. Even though this results in a decrease in the size of its background and interpretation.

Comments on the Quality of English Language

The writing of the manuscript presents some aspects that could be improved, among them cacophony, imprecision and unusual typography. Cacophony is seen in expressions that could improve its readability such as "products, producing" (Line 15), "Aspergillus flavus, Aspergillus oryzae, Aspergillus arachidicola, Aspergillus nomiae, Aspergillus niger, Aspergillus clavatus, Aspergillus nidulans" (Lines 135-137, 287-288, 319-322), "Then .... then" (Lines 165-168), and so on. Some expressions should be checked for accuracy throughout the document; as, for example: the verb "consumed" seems more convenient to refer to "intake"; "have shed light" refers to "elucidated"; "urgent" is "very convenient"; "106 spores per microliter" (Line 206) refers to mL. "29 for 30 m" (line 219) does not refer to meters but to minutes. Aflatoxin B1 requires the number one to be subscript. The expression "aflO" (line 612) requires the letters "afl" to be in italic format. The expression ". And" (line 638) is only correct if preceded by a different symbol (comma or semicolon). Also, the bibliography presents several inconsistencies such as many words of the journals without capitalization or without italic type, spaces between lines with variable widths, etc. For all the above, it is suggested that the complete document be revised to avoid these incorrect expressions that detract from this document, which is of excellent quality.

Author Response

  1. The title of the manuscript (AfHxk1 regulates fungal morphogenesis and pathogenicity of Aspergillus flavus through its domain Hexokinase_2) seems difficult to understand for the lay audience, partly because it starts with a name independent (AfHxk1) from the context of the study. Therefore, it is suggested to reconsider the construction of a simpler title (For example: Regulation of fungal morphogenesis and pathogenicity of Aspergillus flavus through its domain of the Hexokinase AfHxk1).

Answer 1 Thank you, we’ve made corresponding modification to clarify the role of hexokinase according your kind advice, and the title has been changed into “Regulation of fungal morphogenesis and pathogenicity of Aspergillus flavus by Hexokinase AfHxk1 through its domain Hexokinase_2”

2.The abstract does not seem to clearly represent the breadth and depth of the study, as it contains a broad introduction and interpretation, while the methods and results seem to occupy a smaller proportion than expected. It is suggested that the methods and results be presented more fully and schematically. Even though this results in a decrease in the size of its background and interpretation.

Answer 2 Thank you very much! we’ve reduced the proportion of the abstract introduction section and made slight adjustments to the results section in abstract. But in view of the words of the abstract should not exceed 200 as shown below, we could not add too much words into the results section.

Question in the PDF file:

Line 156:Thank you, we have tried to put Fig S1 into the main text Fig.1, but this seems to cause the layout of the picture in each panel to be too small and crowded, and the clarity were obviously reduced. In view of the combination of these two figures would significantly decrease the readers' reading experience, authors tend not to combine these two figures after in-depth discussion.

Reviewer 3 Report

Comments and Suggestions for Authors

·         All abbreviations need to be defined at first use

·         Authors should provide more details on the chromatography aspect, what equipment was used, the parameters, etc., this alone can take a paragraph, I have to question why it was not explicit. Details of the mobile phases should be included as well.

·         How was the TLC done?

·         The “Aflatoxin Analysis” section is too brief

·         The figures are too tiny and difficult to see what the authors are illustrating/ the information they are passing across

·         Figures S1-S5 are not clear and difficult to read

·         Line 196-grammatical error

Comments on the Quality of English Language

I advise the authors to read through the manuscript to correct the minor grammatical errors therein.

Author Response

The writing of the manuscript presents some aspects that could be improved, among them cacophony, imprecision and unusual typography.

Cacophony is seen in expressions that could improve its readability such as "products, producing" (Line 15), "Aspergillus flavus, Aspergillus oryzae, Aspergillus arachidicola, Aspergillus nomiae, Aspergillus niger, Aspergillus clavatus, Aspergillus nidulans" (Lines 135-137, 287-288, 319-322), "Then .... then" (Lines 165-168), and so on.

Answer 1: Thank you, we’ve modified all the expressions based on your kind advice.

Some expressions should be checked for accuracy throughout the document; as, for example: the verb "consumed" seems more convenient to refer to "intake"; "have shed light" refers to "elucidated"; "urgent" is "very convenient"; "106 spores per microliter" (Line 206) refers to mL.

"29℃ for 30 m" (line 219) does not refer to meters but to minutes. Aflatoxin B1 requires the number one to be subscript. The expression "aflO" (line 612) requires the letters "afl" to be in italic format. The expression ". And" (line 638) is only correct if preceded by a different symbol (comma or semicolon).

Answer 2 Thank you, we’ve modified all the expressions based on your kind advice. But because “AflO” refers to protein, therefore we keep the first letter capitalized.

Also, the bibliography presents several inconsistencies such as many words of the journals without capitalization or without italic type, spaces between lines with variable widths, etc. For all the above, it is suggested that the complete document be revised to avoid these incorrect expressions that detract from this document, which is of excellent quality.

Answer 3 Thank you, the format of the reference section has been reviewed and unified according to the requirement of Journal of Fungi.

All abbreviations need to be defined at first use

Answer 4Thank you, we’ve correct the inappropriate abbreviation according to your kind advice.

Authors should provide more details on the chromatography aspect, what equipment was used, the parameters, etc., this alone can take a paragraph, I have to question why it was not explicit. Details of the mobile phases should be included as well.

Answer 5: Thank you, we have written a new paragraph with detail information about the chromatography based on your kind suggestion.

How was the TLC done?

Answer 6: Thank you, we have display the procedure of TLC in the new paragraph begin from Line 221.

The “Aflatoxin Analysis”section is too brief

Answer 7: Thank you, we have provided more details into the "Aflatoxin Analysis" section according to your kind advice.

The figures are too tiny and difficult to see what the authors are illustrating/ the information they are passing across

Figures S1-S5 are not clear and difficult to read

Answer 8Thank you, we’ve we have adjusted the resolution and size of the figures, and

Figure S1-S5 have been adjusted to be clearer.

Line 196-grammatical error

Answer 9: Thank you, the grammatical problem in the corresponding position has been corrected.

Question in the PDF file:

Line 156:Thank you, we have tried to put Fig S1 into the main text Fig.1, but this seems to cause the layout of the picture in each panel to be too small and crowded, and the clarity were obviously reduced. In view of the combination of these two figures would significantly decrease the readers' reading experience, authors tend not to combine these two figures after in-depth discussion.

Round 2

Reviewer 3 Report

Comments and Suggestions for Authors

    .

Comments on the Quality of English Language

Minor edits required